# Climate Change Projections of Terrestrial Primary Productivity over the Hindu Kush Himalayan Forests

Halima Usman[1], Thomas A.M. Pugh[2,3,4], Anders Ahlström[2], Sofia Baig[1*]

[1]Institute of Environmental Sciences & Engineering, National University of Sciences and Technology, Islamabad, 44000, Pakistan

[2]Department of Physical Geography and Ecosystem Science, Lund University, Lund, SE-221 00, Sweden

[3]School of Geography, Earth and Environmental Sciences, University of Birmingham, Edgbaston, Birmingham, B15 2TT, UK

[4]Birmingham Institute of Forest Research, University of Birmingham, Edgbaston, Birmingham, B15 2TT, UK

*Correspondence to*: Sofia Baig (e-mail: sofia.baig@iese.nust.edu.pk)

**Abstract.** Increasing atmospheric carbon dioxide concentration [$CO_2$] caused by anthropogenic activities has triggered a requirement to predict the future impact of [$CO_2$] on forests. The Hindu Kush Himalayan (HKH) region comprises a vast territory including forests, grasslands, farmlands and wetland ecosystems. In this study, the impacts of climate change and land use change on forest carbon fluxes and vegetation productivity are assessed for HKH using the Lund-Potsdam-Jena General Ecosystem Simulator (LPJ-GUESS). LPJ-GUESS simulations were driven by an ensemble of three climate models participating in the CMIP5 (Coupled Model Intercomparison Project Phase 5) database. The modeled estimates of vegetation carbon (VegC) and terrestrial primary productivity were compared with observation-based estimates. Furthermore, we also explored the net biome productivity (NBP) and its components over HKH for the period 1851-2100 under the future climate scenarios RCP2.6 and RCP8.5. A reduced modeled NBP (reduced C sink) is observed from 1986-2015 primarily due to land use change. However, an increase in NBP is predicted under RCP2.6 and RCP8.5. The findings of the study have important implications for management of the HKH region and inform strategic decision making, land use planning and clarify policy concerns.

## 1 Introduction

Anthropogenic activities such as combustion of fossil fuels and land use changes have led to large rises in atmospheric greenhouse gas (GHG) emissions such as carbon dioxide ($CO_2$) and methane over the last century, with atmospheric $CO_2$ mixing ratios increasing from 277 to $409 \pm 0.1$ ppm in 2019 since the preindustrial period, and rising at the mean rate of 2.3 ppm per year from 2010 to 2019 (Friedlingstein et al., 2020) This uptake is likely primarily driven by the fertilizing effects of elevated atmospheric $CO_2$ concentrations on plant growth (Sitch et al., 2015) and by the regrowth of forests following past disturbances (Kondo et al., 2018; Pugh et al., 2019) . However, the ability of this land sink to continue in the future remains highly uncertain (Phillips and Lewis, 2014).

Several studies have identified that warming can cause a stimulation in plant growth by increasing NPP and hence
leading to enhanced carbon uptake (Delpierre et al., 2009;Sullivan et al., 2008;Wu et al., 2011). However, researchers
have also addressed that the rising air temperatures may also stimulate autotrophic respiration in plants (Burton J.
Andrew et al., 2008). Due to global temperature rise, droughts are predicted to increase in frequency, duration and
severity in the future (Trenberth et al., 2013). Increase in temperature causes an exponential rise in vapour pressure
deficit resulting in stomatal closure thus limiting the rate of photosynthesis and higher mortality (Williams et al.,
2013). Hence, the determination of the effect of global rise in temperature on forests is becoming increasingly
important as vegetation response to climate change will result in changes in net carbon uptake, water use efficiency,
plant establishment, carbon biomass allocation and interaction with disturbances (Urban et al., 2017). Several studies
suggest that there is a large gap in the current understanding of the quantification of biomass carbon stock leading to
large uncertainty for the future projections in the ecosystem carbon balance (Ahlström et al., 2012; Jones et al., 2013;
Pugh et al., 2018; Wu et al., 2017).
The HKH region is a diverse and ecological buffer zone, often referred to as the "Third Pole" encompassing an area
of 4.2 million $km^2$. The region provides ecosystem services such as such as watershed protection, livestock shelter
and sustaining communities of estimated 240 million people (Krishnan et al., 2019). The HKH region  has been
experiencing temperature rise of 0.2°C per decade since 1960 (Chen et al., 2013). The forests of HKH are undergoing
changes of varied intensity as a result of climatic and human disturbances, alongside the  various forest management
policies practiced in the different countries (Behera et al., 2018; Pulakesh et al., 2017).  The rate of deforestation along
the HKH has been reported to be 0.5% $yr^{-1}$ in Bhutan and 1.7% $yr^{-1}$ in Myanmar  from 2000 to 2014 (Brandt et al.,
2017). The warming trend observed over recent decades of the $20^{th}$ century is attributed to the increase in
anthropogenic greenhouse gas (GHG) concentrations. The HKH region is believed to be becoming increasingly
sensitive to climate change (Krishnan et al., 2019). In this region, the carbon dynamics are mostly influenced by the
combined effects of climatic change and land-use land-cover change (LULCC) (Almeida et al., 2018;Cao et al., 2018).
Although studies on projections of temperature change exist, but the combined effect of temperature, $CO_2$ and LU
change has not been investigated.
In this paper, the historical and future carbon balance of terrestrial ecosystems in the HKH region are investigated
using results from the Lund-Potsdam-Jena General Ecosystem Simulator (LPJ-GUESS), a DGVM with a detailed
description of forest stand structure and land use (Ahlström et al., 2012; Smith et al., 2001). The goal of the present
study is to (1) evaluate the ability of the LPJ-GUESS model, as forced by climate from a selection of Earth System
Models (ESMs), to reproduce observation-based estimates of vegetation carbon and satellite-derived estimates of
gross primary productivity (GPP) and net primary productivity (NPP) and (2) analyse the spatial and temporal changes
in net biome productivity (NBP) and its components (NPP, Fire and Soil Respiration) and VegC over the period 1851-
70    2100.

## 2 Materials and Methods

### 2.1 Study Area

The HKH region is situated between 16°N–40°S and 61–105°E encompassing Afghanistan, Bangladesh, Bhutan, China, India, Myanmar, Nepal and Pakistan (Figure 1). The evergreen needleleaf forest (ENF) cover about 2.69% of the HKH and 10.5%, 0.06%, 1.09%, 9.37% is covered by evergreen broadleaf forest (EBF), deciduous needleleaf forest (DNF), deciduous broadleaf forest (DBF) and mixed forests (MF) respectively. A major percentage of landcover is covered by open shrublands (OShrub) and grasslands (Grass) occupying 31.57% and 32.08% of the area of HKH. Furthermore, savannas (Sav) and woody savannas (Wsav) cover about 1.19% and 4.46% respectively. The remaining land is covered by croplands (Crop) and closed shrubland (CShrub) with percentage of 5.61% and 1.09% respectively. The forests of the HKH cover about 24% of the region, supporting the 12% of the population of the world by provision of diverse ecosystem goods and ecosystem services including energy, timber and freshwater (Behera et al., 2018)

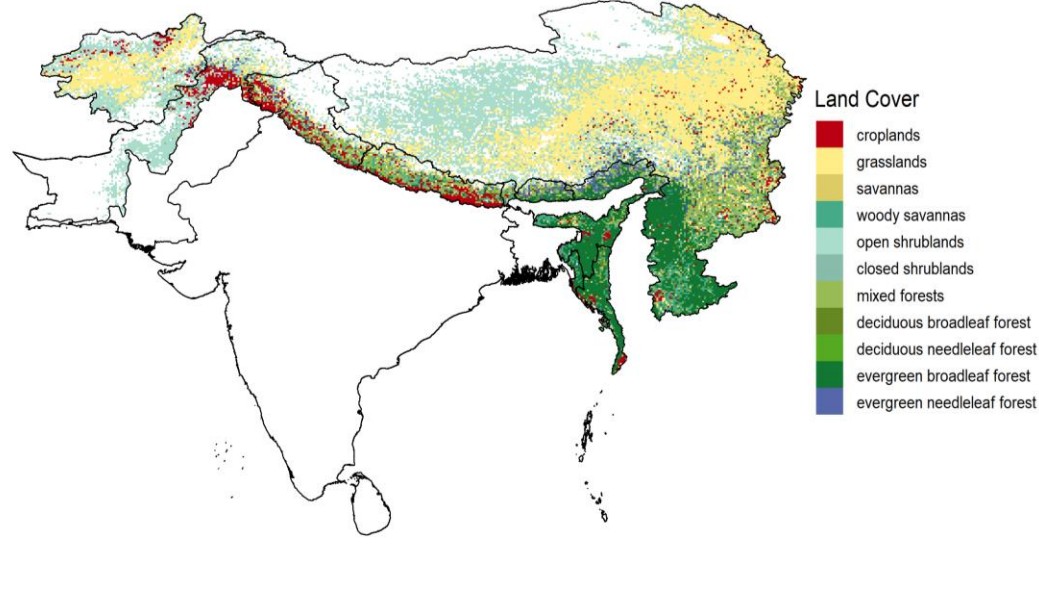

**Figure 1: Land cover of HKH from MODIS (MOD12Q1).**

### 2.2 LPJ-GUESS Ecosystem Model

LPJ-GUESS is a coupled biogeography-biogeochemistry model which integrates process-based representation of terrestrial vegetation dynamics and biogeochemical cycling (Smith et al., 2001). In order to simulate the size of carbon pools in various parts of the plant such as leaves, sapwood, litter and soil the model explicitly accounts for processes such as photosynthesis, allocation and resource competition between plants. The model is useful for predicting the changes in the ecosystem dynamics and is able to simulate and predict the future response of vegetation to elevated $CO_2$ levels at leaf and stand scales (Sitch et al., 2015). In LPJ-GUESS, the species diversity of terrestrial vegetation is represented as groups of species with similar traits known as Plant Functional Types (PFTs). The simulations here

use ten PFTs that are differentiated by attributes such as physiology, morphology, phenology and response to
disturbance along with bioclimatic constraints. Trees are modelled as age cohorts across multiple replicate patches,
but are identical within each cohort (age class) (Smith et al., 2001).

LPJ-GUESS works on a daily time steps, with some processes, such as vegetation dynamics, computed annually. The
input data to the model includes atmospheric $[CO_2]$ mixing ratio, precipitation, shortwave radiation, air temperature
and soil type. Simulations begin from bare ground, and go through a 500 year "spin-up phase" during which soil and
carbon litter pools accumulate and reach a state of equilibrium. An analytical solution is used to accelerate spin-up of
the soil carbon pools. In the spin-up phase the model is forced by constant $[CO_2]$ and a repeated detrended 30-year
climate segment from the beginning of the climate dataset used. As the spin-up phase finishes, the "transient phase"
begins, in which land use, climate and $[CO_2]$ evolve over time as specified in the forcing datasets. Here we analyse
outputs of vegetation carbon, gross primary productivity, net primary productivity and net biome productivity and its
components.

**2.3 Simulation Protocol**
In this study simulations are reanalysed from (Ahlström et al., 2012) with a focus on the HKH region. Only an
overview of the salient features of the set-up are given for this study. For more set-up details, please see Ahlström et
al., (2012). Spatial patterns of carbon pool, fluxes and terrestrial primary productivity were investigated in HKH
forests by using the output simulations of LPJ-GUESS resolution of $0.5° \times 0.5°$ with climate forcing from climate
models participating in CMIP5 (Table 1) under RCP 2.6 (Van Vuuren et al., 2007) and RCP8.5 representative
concentration pathway (Riahi et al., 2011). RCP2.6 emission pathway is representative of reduced GHG concentration
levels. It is a defined as a "peak-and-decline" scenario, in which the radiative forcing level first reaches around 3.1
$W/m^2$ by mid-century, and return to a value of 2.6 $W/m^2$ by 2100. In contrast, RCP8.5 is characterized by increasing
GHG emissions over time, culminating in a radiative forcing of 8.5 $W/m^2$ in 2100. The climatic data was bias corrected
by using CRU TS 3.0 (Mitchell and Jones, 2005) 1961-90 climatologies on annual and monthly basis (seasonal bias
correction). The monthly fields of precipitation, downward shortwave radiation and air temperature were bi-linearly
interpolated to the CRU grid at a resolution of $0.5° \times 0.5°$. The correction by the climatology fields (1961-90) adjust
for bias in annual averages and seasonal distribution. Figure S1 (a) and S1 (b) shows an example of how bias correction
adjusts the time series of temperature and precipitation.

Croplands and pastures were treated as natural grasslands in the vegetation model in simulations that simulated land
use (LU) (Ahlström et al., 2012). To assess the impact of human land use, simulations containing potential natural
vegetation (PNV) were also assessed in comparison to those containing LU for both RCP2.6 and RCP8.5.





| Modelling Center | Institute ID | Model name |
|---|---|---|
| National Center for Atmospheric Research | NCAR | CCSM4 |
| Institut Pierre–Simon Laplace | IPSL | IPSL-CM5A-MR |
| Max Planck Institute for Meteorology | MPI-M | MPI-ESM-LR |


**Table 1: CMIP5 models and modelling groups used to provide climate forcing data for LPJ-GUESS in this**
**study.**

**2.4 Model Evaluation**
In this study, a global dataset of forest above-ground biomass (AGB) developed within European Commission-funded
GEOCARBON project was considered for the purpose of comparison with LPJ-GUESS VegC. The base year of this
dataset is 2000. As LPJ-GUESS VegC includes both above- and below-ground vegetation carbon, the AGB of
GEOCARBON was converted into VegC by applying a correction to estimate below-ground biomass in the
GEOCARBON dataset based on (Saatchi et al., 2011). The resulting above and below ground biomass was converted
to carbon content by multiplying by 0.5.

Furthermore, the Moderate-resolution Imaging Spectroradiometer (MODIS) GPP and NPP product (MOD17A3H)
was used for comparison with the modelled GPP and NPP. MOD17 is based on the light use efficiency approach and
consists of two products, MOD17A2 and MOD17A3 (Zhao and Running, 2010). In this study we incorporated
MOD17A3 that contains annual sums of GPP and NPP with a 0.0083∘ × 0.0083∘ spatial resolution for the period
2000–2010. In order to compare LPJ-GUESS GPP and NPP estimates, MOD17A3 GPP and NPP datasets were
downloaded from "The Application for Extracting and Exploring Analysis Ready Samples (AρρEEARS)" website
("LP DAAC - AppEEARS".). Land cover (MOD12Q1) used in this study was downloaded from
files.ntsg.umt.edu/data/NTSG_Products/MOD17/GeoTIFF/MOD12Q1/ and was used for land cover stratification
(Friedl et al., 2002). Land cover related to barren, water and urban were masked from LPJ-GUESS data in order to
make it comparable with MOD17A3 data (i.e. identical spatial extent, land cover classes and number of grid cells).
Both GEOCARBON and MODIS datasets were aggregated to 0.5° x 0.5° resolution for comparison with LPJ-GUESS.

## 3 Results

### 3.1 Comparison between Observed and LPJ-GUESS estimations of VegC

Simulations forced by three CMIP5 ESMs of mean VegC from 1986-2015 were compared with the observed GEOCARBON dataset (Figure 2). The mean VegC of observed dataset was estimated to be 4.68 kg C m$^{-2}$. While the modeled VegC for HKH averages 1.93 kg C m$^{-2}$, 2.04 kg C m$^{-2}$ and 2.14 kg C m$^{-2}$ for simulations forced by climate outputs from IPSL-CM5A-MR, MPI-ESM-LR and CCSM4 respectively. Most of the difference is found to be the southern regions of HKH. A moderate agreement was found between the GEOCARBON and LPJ-GUESS VegC with a mean r$^2$ value of 0.44.

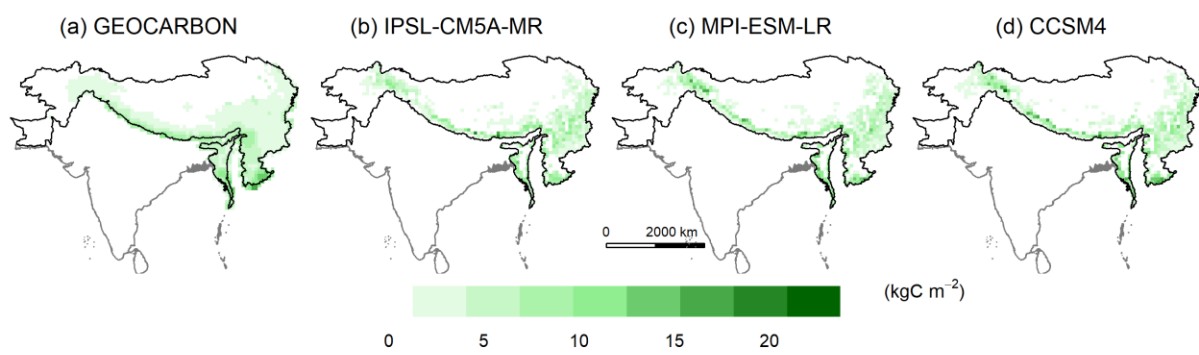

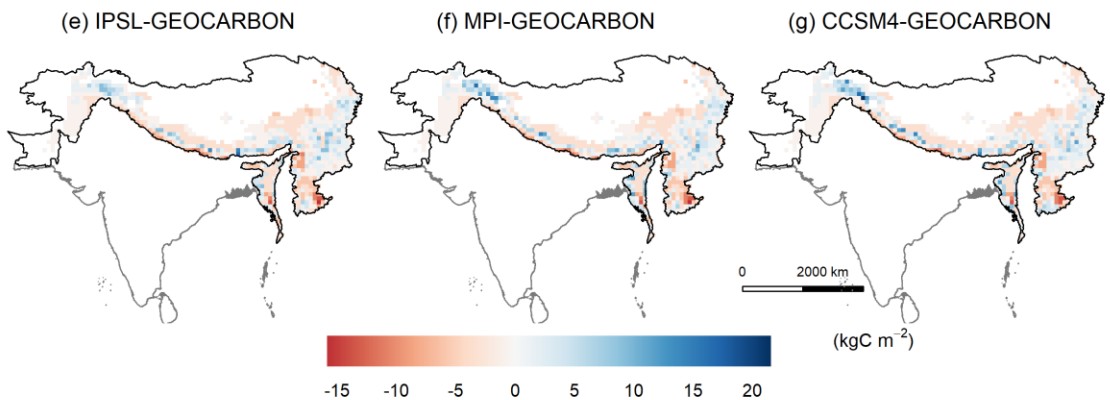

**Figure 2: The distribution of VegC as simulated by (a) GEOCARBON, (b) IPSL-CM5A-MR, (c) MPI-ESM-LR (d) CCSM4 and (e,f,g) their respective differences with GEOCARBON dataset for the HKH region.**

Furthermore the simulations of the CMIP5 models and the observed estimations in the HKH region were compared according to land cover classes from MOD12Q1 (Figure 3). There is an underestimation of VegC in evergreen broadleaf forests. The mean GEOCARBON VegC was 7.73 kg C m$^{-2}$ was on average, 2.68 kg C m$^{-2}$ higher than LPJ-GUESS VegC for evergreen broadleaf forest. VegC for remaining forest types showed a lesser difference than 1.5 kg

C m$^{-2}$.  The simulation of VegC was not very sensitive to differences in the bias-corrected modelled climates from the
CMIP5 models for the period from 1986-2015.

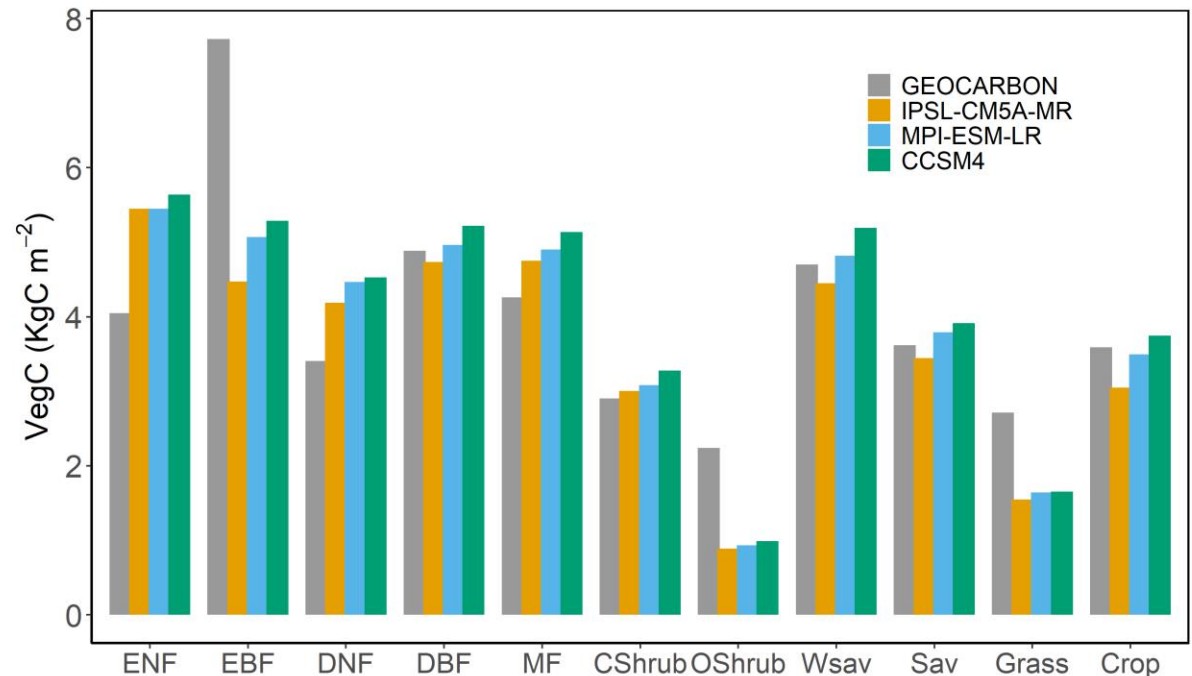


**Figure 3: Summary statistics of LPJ-GUESS and GEOCARBON VegC for HKH in KgC m$^{-2}$ of CMIP5**
**models according to land cover classes**



**3.2 Evaluation of patterns of GPP and NPP from 2000-2010**
The mean MODIS GPP for 2000-2010 was estimated to be $0.69 \pm 0.26$ kgC m$^{-2}$ yr$^{-2}$. The GPP for IPSL-CM5A-MR,
MPI-ESM-LR and CCSM4 was $0.84 \pm 0.17$ kgC m$^{-2}$ yr$^{-1}$, $0.83 \pm 0.16$ kgC m$^{-2}$ yr$^{-1}$ and $0.88 \pm 0.16$ kgC m$^{-2}$ yr$^{-1}$
respectively (Figure 4). The mean MODIS NPP was estimated to be $0.38 \pm 0.12$  kgC m$^{-2}$ yr$^{-1}$ and $0.43 \pm 0.07$ kgC m$^{-}$
$^2$ yr$^{-1}$, $0.42 \pm 0.07$ kgC m$^{-2}$ yr$^{-1}$, and $0.44 \pm 0.07$ kgC m$^{-2}$ yr$^{-1}$ for IPSL-CM5A-MR, MPI-ESM-LR and CCSM4
respectively (Figure 4). Both of the spatial datasets are able to capture important features such as the low productive
Himalayan barren areas in the north and high productive regions like the forests and croplands in lower parts of HKH
region (Figure S2 & S3). There was a moderate spatial agreement between the MODIS and modelled GPP with mean
r$^2$ values of 0.54. However, there was a weaker correlation between the satellite-derived and modelled NPP with mean
r$^2$ values of 0.4. Averaged GPP and NPP from MODIS and LPJ-GUESS per land cover classes from MOD12Q1 are
shown in figure 5(a) and 5(b) respectively. A difference is found in the EBF land cover class when both datasets are
compared. GPP for MODIS was estimated to be 2.48 kgC m$^{-2}$ yr$^{-1}$ and for average ESMs GPP was estimated to be
1.34 kgC m$^{-2}$ yr$^{-1}$. Furthermore MODIS NPP was estimated to be 1.26 kgC m$^{-2}$ yr$^{-1}$ and the ESMs average NPP was
0.56 kgC m$^{-2}$ yr$^{-1}$.

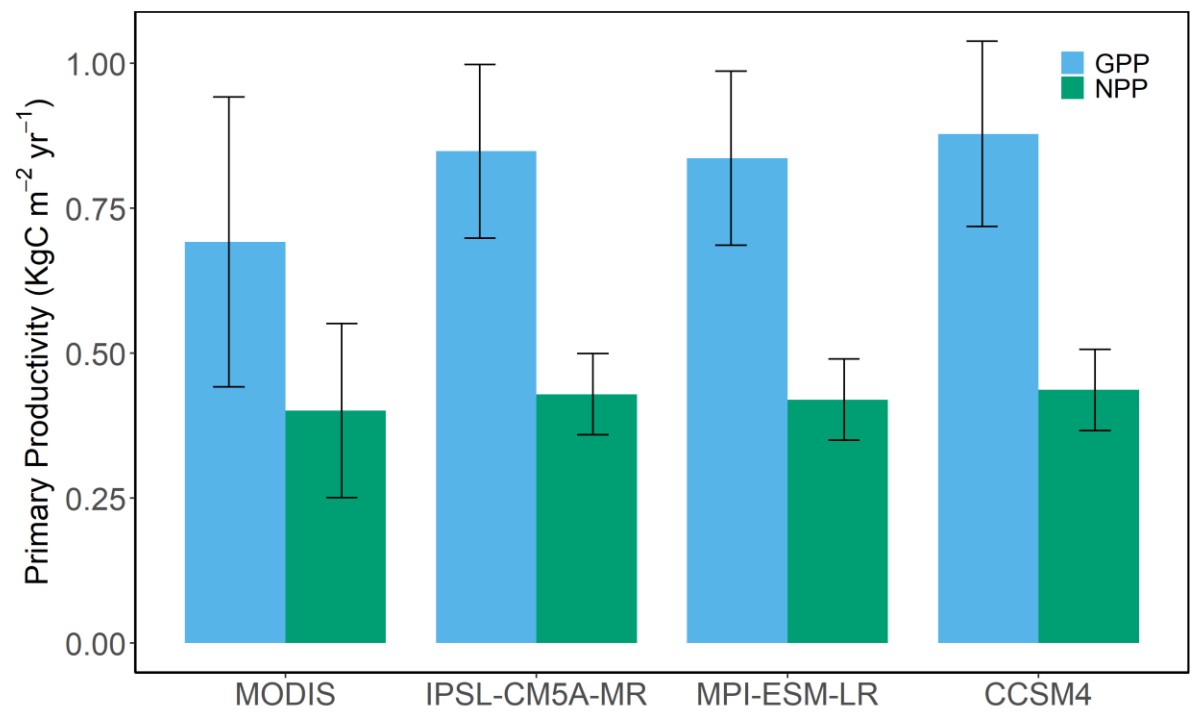


**Figure 4: GPP and NPP for HKH showing mean GPP (blue) and mean NPP (green) from MOD17 and from**
**the LPJ-GUESS model forced by climate outputs from the 3 ESMs (average for the period 2000–2010).**
**Vertical black bars illustrate ± standard error where *n*=11**

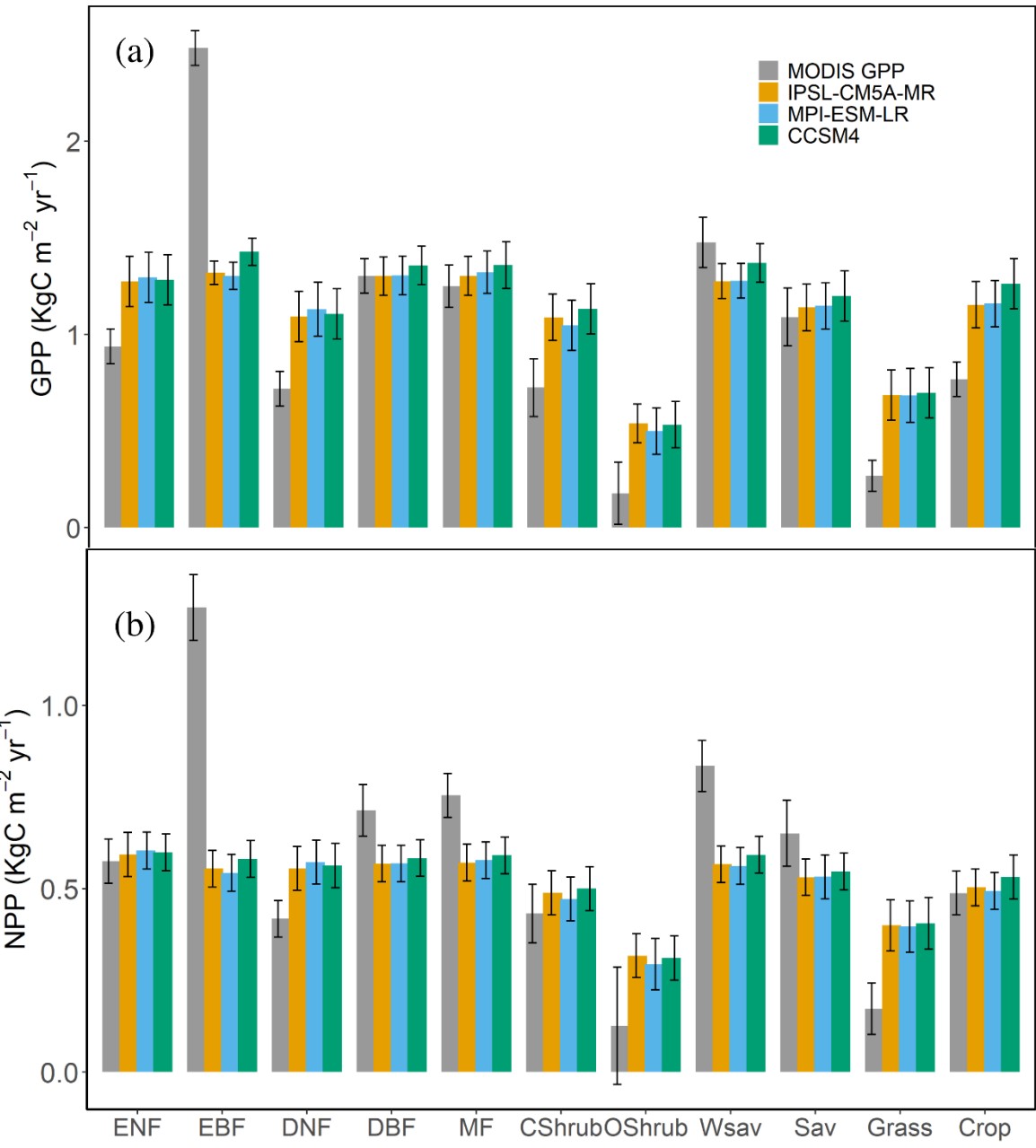

**Figure 5: (a) Mean MOD17 and LPJ-GUESS GPP per land cover class (b) Mean MOD17 and LPJ-GUESS NPP per land cover class. Vertical black bars illustrate ± standard error where *n*=11.**



**3.3 Evaluation PFTs distribution in LPJ-GUESS**


Figure 6 shows the distribution of the PFT simulated by the LPJ model in the HKH region. The LPJ-GUESS PFT
distribution was compared to the land cover classes of MOD12Q1 dataset. A major part of C3 grasses (C3G) found
in in the majority of HKH area including Tibetan Plateau and West pats of the HKH region. MOD12Q1 classifies this
area as open shrublands and grasslands, which is consistent given that shrubs are not explicitly included with the ten
global PFTs used. The modelled data and observed data correspond well to each other in terms of the major features
of the broadleaf forests. In LPJ-GUESS, regions of Bangladesh and Myanmar, most of the area is covered by tropical
broadleaf raingreen forests (TrBR), whereas MOD12Q1 land cover classification shows those areas to be classified
as evergreen broadleaf forests. There was minimal difference in to 2000-2010 PFT distribution between the three
ESMs climates.

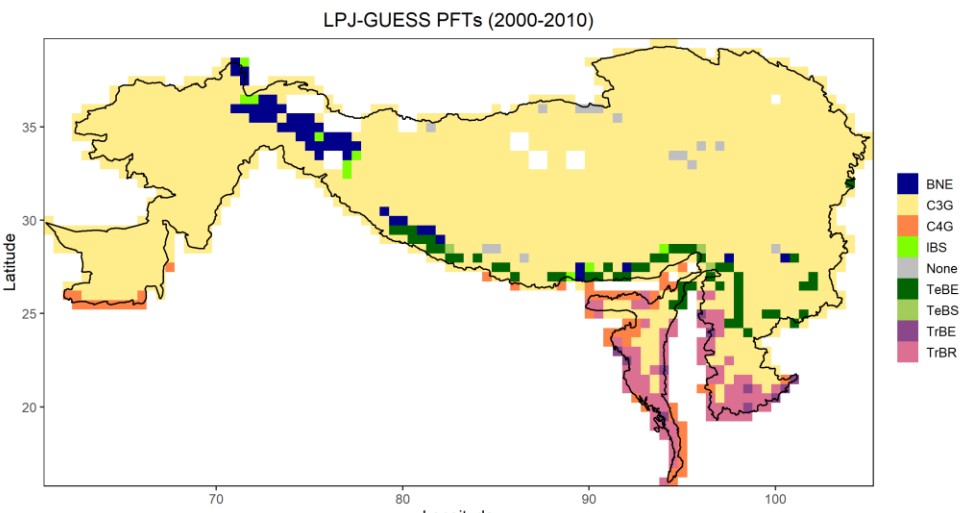

**Figure 6: Average distribution of PNV simulated from 2000-2010 by LPJ-GUESS forced by CCSM4 climate.**
**Full PFT names (as shown in legend): BNE = boreal needle-leaved evergreen tree; C3G = C3 grass; C4G = C4**
**grass; IBS = shade intolerant broadleaved; TeBE = temperate broadleaved evergreen tree; TeBS = temperate**
**broad-leaved summergreen tree; TrBE = tropical broad-leaved evergreen tree; TrBR = tropical broadleaved**
**raingreen tree**

**3.4 Projected Spatial Changes in the Pattern of NBP and Components**
Two types of simulations were used in order to make a comparison to assess the spatial patterns of NBP. The
simulations derived from the potential natural vegetation (PNV) were compared with simulations from land use (LU)
simulations generated by LPJ-GUESS model.  NBP changes with PNV and LU were calculated for three time periods
of past period (1851-1880), present period (1986-2015) and RCP2.6 and RCP8.5 representing the future scenario from
2071 to 2100. In PNV simulations for 1851-1880, the mean NBP for the three ESM climates was estimated to be
0.003 kgC m$^{-2}$ yr$^{-1}$. It increased to 0.037 kgC m$^{-2}$ yr$^{-1}$ in 1986-2015. For RCP2.6 and RCP8.5, in the LU simulations,
the NBP increases to 0.015 kgC m$^{-2}$ yr$^{-1}$ and 0.04 kgC m$^{-2}$ yr$^{-1}$, showing a dampening effect of land-use change on
NBP increases. The simulations show a shift from carbon source to sink in both future scenarios in both simulations,
with higher NBP in RCP8.5 compared to RCP2.6. Most of the carbon sink in the future scenarios is seen in central
and lower region of HKH (Fig. S4). The Tibetan Plateau acts as a carbon sink as warming temperature and carbon
fertilisation stimulate vegetation growth in the future RCP8.5 scenario.

NBP was broken down into its component fluxes of NPP, Fire and Soil Respiration rate (Figs. S5-7). Simulations of
average NPP in the PNV and LU simulations in the past period (1851-1880) reached on average 0.306 kgC m$^{-2}$ yr$^{-1}$
and 0.303 kgC m$^{-2}$ yr$^{-1}$ respectively. The present day mean NPP across HKH was estimated to be 0.388 kgC m$^{-2}$ yr$^{-1}$
and 0.377 kgC m$^{-2}$ yr$^{-1}$ for PNV and LU simulations respectively. The simulated NPP increased to 0.452 kgC m$^{-2}$ yr$^{-1}$
in PNV simulations and 0.437 kgC m$^{-2}$ yr$^{-1}$ in the LU simulations in RCP2.6. Furthermore in RCP8.5 the NPP
increased to 0.657 kgC m$^{-2}$ yr$^{-1}$ in PNV simulations and 0.622 kgC m$^{-2}$ yr$^{-1}$ in the LU simulations. Human land use
thus moderately reduced future increased in NPP. An average value of fire flux was estimated to be 0.065 kgC m$^{-2}$
yr$^{-1}$ and 0.041 kgC m$^{-2}$ yr$^{-1}$ by LPJ-GUESS for the past period for PNV and LU simulations respectively. In the
present period, the model simulates a slightly higher average fire flux of 0.065 kgC m$^{-2}$ yr$^{-1}$ in PNV simulations,
compared to 0.042 kgC m$^{-2}$ yr$^{-1}$ in LU simulations. For future scenario, it is predicted that in the RCP2.6 the fire flux
will increase with an estimated value of 0.08 kgC m$^{-2}$ yr$^{-1}$ and 0.046 kgC m$^{-2}$ yr$^{-1}$ for PNV and LU simulations
respectively. The lower fire fluxes in the LU scenarios reflect the large area of land dedicated to agriculture, which
increases over time. Agricultural land is assumed not to contribute to fire fluxes in these simulations. In future scenario
RCP 8.5 it is predicted that the fire flux will increase to a mean of 0.081 kgC m$^{-2}$ yr$^{-1}$ in HKH. In PNV simulated soil
respiration, an overall increasing trend is seen in the HKH region. In PNV simulated soil respiration, an overall
increasing trend is seen in the HKH region. A lower rate of soil respiration is projected in the future scenario, with a
mean value of 0.053 yr$^{-1}$ and 0.054 yr$^{-1}$ in RCP2.6 for PNV and LU simulations respectively. For RCP8.5, the mean
soil respiration rate was found to be 0.075 yr$^{-1}$ for both PNV and LU simulations.

Table S8 shows the average projected changes in NBP, NPP, Fire and Soil respiration rate forced by LPJ-GUESS by
climate outputs from the 3 ESM climates for past period (1851-1880), present period (1986-2015) and future scenario
(2071-2100) under RCP2.6 and RCP8.5. The choice of ESM climate had a minor effect on the results.







**3.5 Projected Temporal Changes in the Pattern of NBP and Components according to Elevation**

Most of the high elevation region including the Tibetan Plateau Region is devoid of forest area as it experiences a mean annual temperature of less than -2°C. Hence the area below 4500 m is classified as low elevation and elevation above 4500 m is classified as high elevation (Pulakesh et al., 2017). Figure 7(a-d) , summarizes the temporal patterns of NBP, NPP, Fire and soil respiration according to low elevation and high elevation. In the past period from 1851-1880, the NBP flux is positive in lower elevation regions (0-4500 m) of HKH as compared to higher elevation areas. The HKH region was a carbon source in the period from 1851-1880; sink strength at elevation 0 to 4500 m increased from 1986 onwards, resulting in a carbon sink, and it became a relatively strong sink in the future scenario in RCP8.5. In RCP8.5, the PNV simulations estimated a NBP of 0.02 kgC $m^{-2}$ $yr^{-1}$ and in LU simulation it was estimated to be 0.01 kgC $m^{-2}$ $yr^{-1}$. However at higher elevation in PNV simulations, the NBP was estimated to be 0.12 kgC $m^{-2}$ $yr^{-1}$ and 0.08 kgC $m^{-2}$ $yr^{-1}$ in LU simulations.

We also analysed the change in NPP during the period from 1851 to 2100 and found that there was an upward trend in both lower and higher elevation in simulations including PNV and LU simulations. PNV simulated NPP is projected to increase from 0.31 kgC $m^{-2}$ $yr^{-1}$ to 0.39 kgC $m^{-2}$ $yr^{-1}$ from 1851-1880 and 1986-2015. In future scenario for PNV simulations the NPP is estimated to be 0.46 kgC $m^{-2}$ $yr^{-1}$ in RCP2.6 and 0.66 kgC $m^{-2}$ $yr^{-1}$ RCP8.5 respectively. For LU simulations the NPP is projected to increase from 0.31 kgC $m^{-2}$ $yr^{-1}$ to 0.38 kgC $m^{-2}$ $yr^{-1}$ from 1851-1880 and 1986-2015 respectively. In future scenario, NPP in RCP2.6 is estimated to be 0.44 kgC $m^{-2}$ $yr^{-1}$ and 0.63 kgC $m^{-2}$ $yr^{-1}$ in RCP8.5 in LU simulations.

The temporal trend of fire flux from 1851-2100, showing generally higher flux values in PNV simulations as compared to LU simulations. At lower and higher elevations, an increasing trend of fire flux is seen. A higher fire flux is projected in the RCP8.5 scenario with a mean value of 5.9 kgC $m^{-2}$ $yr^{-1}$ and 7.08 kgC $m^{-2}$ $yr^{-1}$ in both PNV and land use simulations respectively. The rate of soil respiration shows an increasing trend from the period of 1851-2100. A higher soil respiration rate is projected in higher elevation in RCP8.5 compared to RCP2.6 in PNV model simulations and LU model simulations. A similar trend was found in the climatic model MPI-ESM-LR included in the supplementary information (Figure S9).

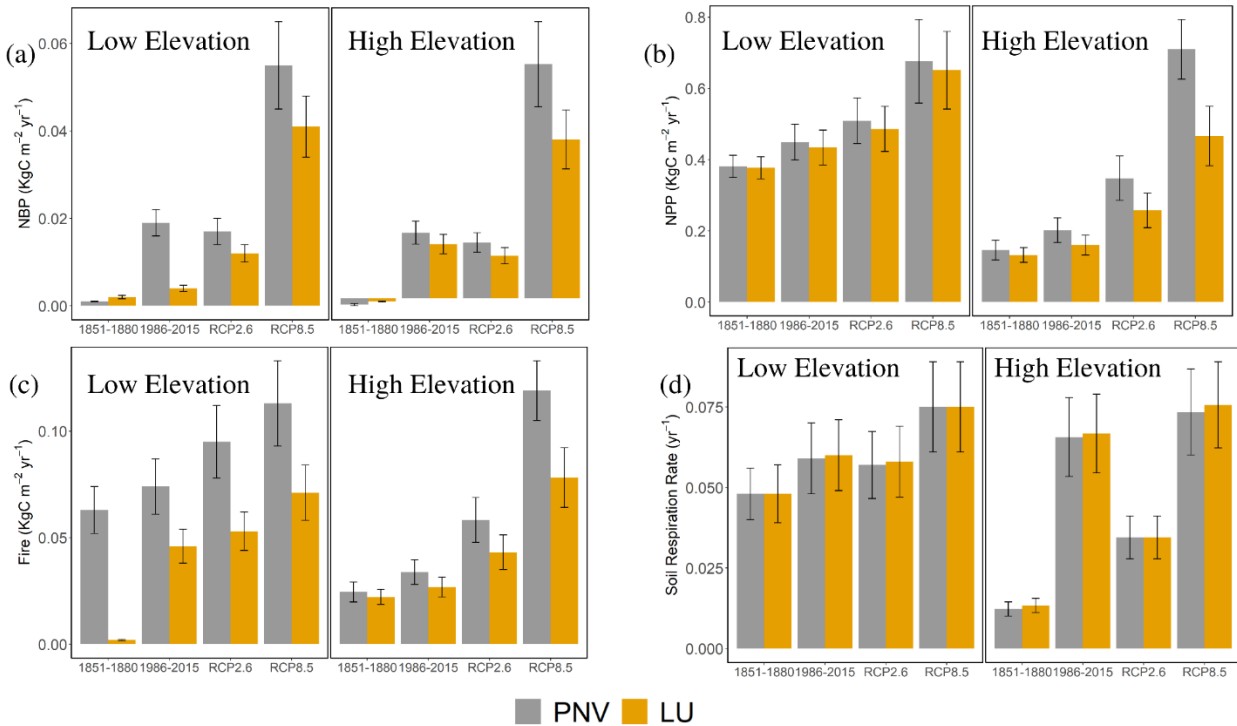


**Figure 7 LPJ-GUESS simulated distribution by CCSM4 on a)NBP b) NPP c) Fire d) Soil Respiration rate in**
**HKH according lower elevation (0-4500 m)and higher elevation (greater than 4500m) for PNV (grey color)**
**and land use change (orange color). Vertical black bars illustrate ± standard error where *n*=30**




**3.6 Projected Spatial Changes in the Pattern of Vegetation Carbon**

Model estimates of VegC in HKH terrestrial ecosystems have increased since 1986 and will increase under both future
climate scenarios in both PNV and LU simulations. For simulations with no land use, the mean VegC is estimated to
be 3.58 kg C m$^{-2}$, 4.05 kg C m$^{-2}$ for past and present period and is projected to reach to 5.51 kg C m$^{-2}$ and 7.19 kg C
m$^{-2}$ under RCP2.6 and RCP8.5 respectively. Furthermore, for the LU simulations, the VegC is estimated to be 2.95
kg C m$^{-2}$ in the past period and slightly decreasing to 2.14 kg C m$^{-2}$ in the present period. An increase in VegC is
predicted in both scenarios, with a mean value of 2.45 kg C m$^{-2}$ and 3.80 kg C m$^{-2}$ for RCP2.6 and RCP8.5 respectively.
Spatial patterns show that the mean VegC (Figure 8) will increase most in the lower belt of the HKH region and north
eastern region in HKH during 2071-2100 under both the RCP2.6 and RCP8.5 scenarios.



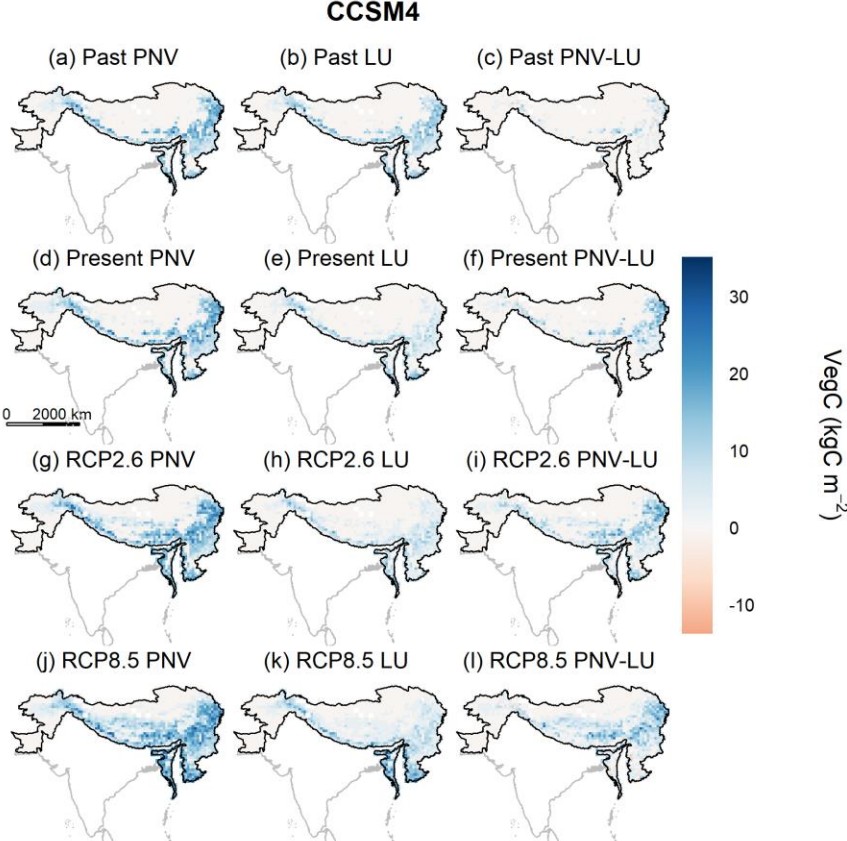

**CCSM4**

(a) Past PNV  (b) Past LU  (c) Past PNV-LU

(d) Present PNV  (e) Present LU  (f) Present PNV-LU

(g) RCP2.6 PNV  (h) RCP2.6 LU  (i) RCP2.6 PNV-LU

(j) RCP8.5 PNV  (k) RCP8.5 LU  (l) RCP8.5 PNV-LU


311          **Figure 8 LPJ-GUESS simulated distribution by CCSM4 on VegC in HKH region under a) past period (1851-**
**1880) with PNV b) past period (1851-1880) with land use change c) difference between past PNV and past LU d) present period**
**(1986-2015) with PNV e) present period (1986-2015) with land use change f) difference between present PNV and past LU g)**
**future scenario RCP2.6 (2071-2100) with PNV h) future scenario RCP2.6 with LU (2071-2100) i) difference between future**
**RCP2.6 PNV and LU j) future scenario RCP8.5 (2071-2100) with PNV k) future scenario RCP8.5 with LU l) difference between**
**future RC8.5 PNV and LU**


**3.7 Comparison of observational climate products**

Figure 9 and Figure 10 shows a comparison between CRU and ERA5 datasets of temperature and precipitation from
1979 to 1990 respectively. The mean CRU temperature from 1979 to 1990 was estimated to be 5.64° C and for ERA5
it was estimated to be 4.32° C. Both of the datasets capture higher temperature in the lower region of the HKH, with
warmer temperature in Bangladesh and Myanmar. On the other hand low temperature are observed in the region of
Tibetan Plateau, The two datasets overall showed a strong agreement with a strong correlation of 0.96. However, the
agreement of spatial distribution of precipitation showed a lower correlation with an $r$ value of 0.67. There is a
difference of mean precipitation in lower region of eastern HKH. CRU dataset, shows an average precipitation of
0.0018 m day$^{-1}$, whereas ERA5 data shows an estimation of 0.0028 m day$^{-1}$.

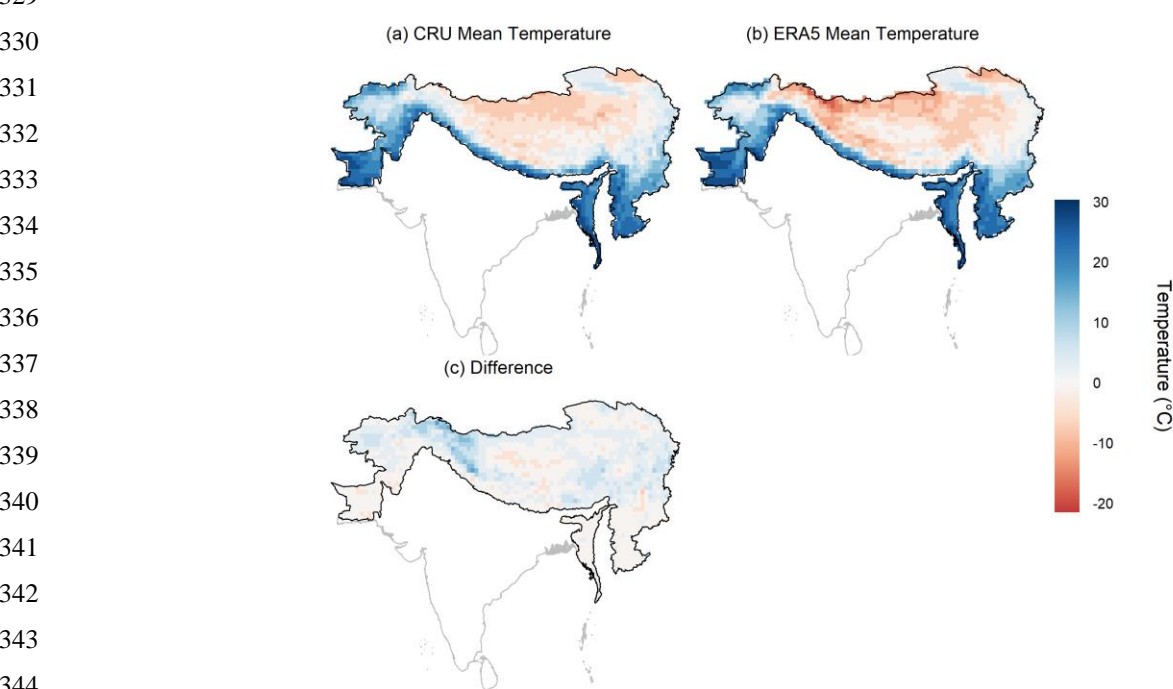

(a) CRU Mean Temperature

(b) ERA5 Mean Temperature

(c) Difference

**Figure 9 Comparison of temperature (a) average CRU (1979-1990) (b) ERA5 data (1979-1990) (c) and the difference**
**between ERA5 and CRU dataset in degree Celsius**

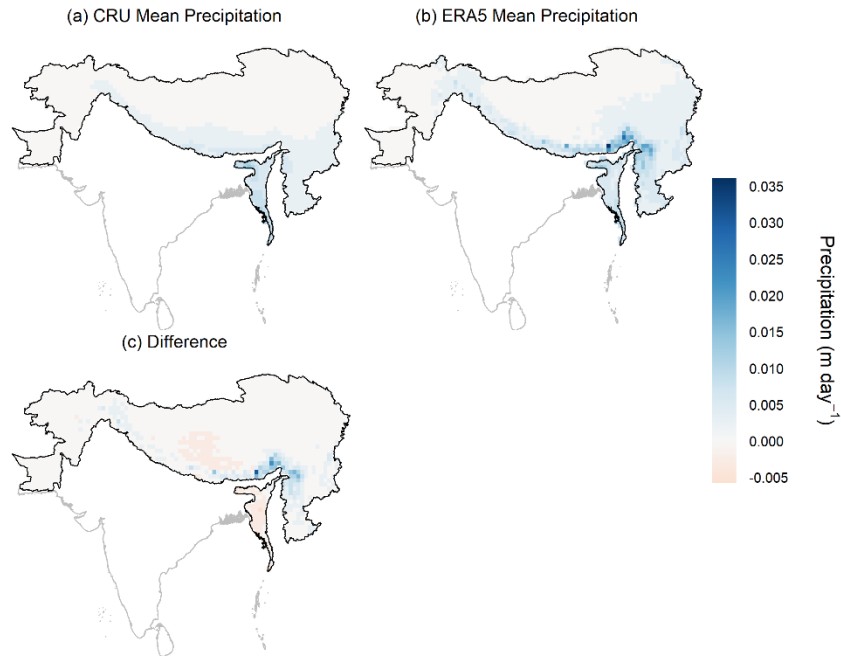

(a) CRU Mean Precipitation

(b) ERA5 Mean Precipitation

(c) Difference


**Figure 10 Comparison of precipitation (a) average CRU (1960-1990) (b) ERA5 data (1979-1990) (c) and the difference**
**between ERA5 and CRU dataset in m day$^{-1}$**

**4 Discussion**

We compared the modelled simulations VegC and primary productivity with satellite-based estimates. For VegC, the comparator dataset is a global aboveground biomass map from the GEOCARBON project for the year 2000. A good agreement was found between GEOCARBON and the ESMs with relatively little difference between the ESM climates. The difference between modelled and observed VegC was found in the EBF and may be attributed due to the differences in terms of the coverage of aboveground or belowground biomass of both datasets. The GEOCARBON dataset includes the spatial distribution of forest biomass covering only the aboveground vegetation for 2000. On the other hand, LPJ-GUESS simulation cover both above and belowground. Hence uncertainties may rise due to the converting aboveground biomass to the total of aboveground and belowground biomass for the datasets of GEOCARBON on order to be comparable with LPJ-GUESS VegC. Furthermore the satellite-derived biomass dataset GEOCARBON was generated by harmonization of datasets of two different years. The tropical biomass products represent the year 2000 status of forests, and the boreal aboveground biomass maps are based on spaceborne radar data from the year 2010. The LPJ-GUESS VegC was averaged over the years from 1986 to 2015. Hence the difference in the years of observations might have introduced additional uncertainty. This drawback of observed dataset was also highlighted by Li et al. (2017).

Secondly, we compared the LPJ-GUESS GPP and NPP with MODIS datasets from 2000-2010. A higher GPP and NPP emerged in areas covered with dense forests mainly in the southeast and southwest HKH region, especially in Bangladesh and Myanmar. The LPJ-GUESS GPP showed a better agreement with GPP MODIS than NPP MODIS. It is important to note that the LPJ-GUESS simulations here and the MODIS algorithm do not share common meteorological drivers and that might reduce the correlation between the two datasets (Liu et al., 2018). Previous studies have also reported that DGVMs generally overestimate GPP in the Northern Hemisphere (Li et al., 2016). This could be attributed to the absence of parametrization of tropospheric ozone that leads to overestimation of LAI leading to increased GPP (Anav et al., 2013). Yet most of the researchers suggest that simulated GPP by DGVMs were neither overestimated nor underestimated, but the results are limited by number of observational or model considerations. For instance, the modelled LPJ-GUESS simulations here do not include the impact of nitrogen cycling (Li et al., 2016). The inconsistencies of primary productivity for EBF were also observed in various studies (Ardö, 2015; Garrigues et al., 2008). Study carried out by Ardö (2015), estimated MOD17 GPP to be 0.8 kgC m$^{-2}$ higher compared to LPJ-GUESS GPP for EBF land cover class. Areas affected by frequent cloud cover or atmospheric contamination may then show inconsistent estimates of vegetation productivity using MOD17.

The second step was to explore the variability of NBP and its components and VegC over HKH from 1851-2100 with PNV and LU simulations and how this variability was influenced by elevation. Results showed that the terrestrial ecosystems of HKH had been a carbon sink for the period of 1851-2015 with a generally positive NBP and the region is projected to remain a carbon sink in both future scenarios. However in the simulations containing land use, the sink strength of the region is lower than in the potential natural simulations. Past modelling studies (Houghton et al., 1987) did capture a large net release of carbon in the 1980s from Nepal, Bangladesh, Bhutan, India, Pakistan, Myanmar and

China due to land use change mainly deforestation. Extensive research has shed light on the serious degradation of
grasslands on the Tibetan Plateau of China due to anthropogenic disturbances since the 1960s (Joshi et al., 2013;Wang
et al., 2008). This degradation appears to be captured well by the LPJ-GUESS simulation as a reduction of NBP in
parts of China can be seen in the spatial maps from 1986-2015. Furthermore, a  recent study carried out by (Calle et
al., 2016) calculated the regional carbon fluxes LULCC in Asia for the period from 1980 to 2009, using eight carbon
cycle DGVMs. Since the 1980s, the ensemble mean of the DGVMs also have shown a net carbon source from South
Asian and East Asian land ecosystems. From 1951 to 2005, most parts of the HKH underwent rapid population and
economic growth increasing the demand for natural resources, hence resulting in large changes in LULCC and habitat
fragmentation.

The LPJ-GUESS simulations for the HKH for 2071-2100 for both scenarios predicted a net sink of carbon. The
simulations of LPJ-GUESS of HKH region was consistent with the previous studies carried out at a global scale where
a C sink was reported in the future scenario by various DGVMs during the next century (Cramer et al., 2001).  A
greater increase in NBP and VegC was seen in RCP8.5, as the rate of photosynthesis by terrestrial vegetation rises
due to increase with atmospheric $CO_2$ content leading to increased carbon uptake. A global scale study carried out by
(Thompson et al., 2004) discussed that the $CO_2$ fertilization could limit the global warming in the future scenario,
however the nutrient limitations, which were not considered here, could weaken this effect. The influence of carbon-
nitrogen interactions has a greater effect in the colder climates as compared to carbon only interactions due to inability
of newly established vegetation to compete for the nitrogen resources with existing vegetation under nitrogen
limitation (Wärlind et al., 2014) . However, the version of LPJ-GUESS used in this study did not take account of
nutrient limitations and assume nitrogen to be at an optimal level for the terrestrial vegetation. The coupling of carbon
and nitrogen cycles are becoming widely recognized as nitrogen dynamics have been incorporated into global C
cycling model (Fleischer et al., 2015).

In this study, the NPP increased from the period of 1851 to 2100. A higher NPP was simulated in RCP8.5, as increasing
temperature and $CO_2$ concentration level leads to increased NPP (Azhdari et al., 2020). The dominant fire occurrences
taking place in HKH region are savanna fires that includes grasslands fires and fires caused by deforestation and forest
degradation (Van Der Werf et al., 2010).   The ESMs used to force LPJ-GUESS simulated temperature and
concentration $CO_2$ levels (Figure S10) in RCP2.6 and RCP8.5 steadily increases from 2000 onwards. Hence with
rising temperatures, the loss of carbon due to biomass burning in wildfires cause the drier forests to become more
vulnerable to climate change as they are more sensitive to fire and droughts (Anderson-Teixeira et al., 2013). Studies
of DGVMs indicate that in the absence of land use changes (Sitch et al., 2015), the soil respiration rate increases with
climate change, however the simulations in this study taking account of land use changes have also shown an increase
in soil respiration rate. Climatic warming is considered to stimulate the rates of soil respiration, potentially resulting
in further increases in global temperatures by accelerating the rate of carbon feedback cycle via $R_a$ and decomposition
of organic matter (Carey et al., 2016).

The study also assessed the comparison of observational climate products over HKH for the period 1979-1990. Our
analysis for precipitation showed that the ERA5 climatic data has higher precipitation of 0.009 m day$^{-1}$ in the HKH
region of the evergreen broadleaf forests. However for CRU climatic dataset the precipitation was estimated to be
0.005 m day$^{-1}$. Hence the underestimation in primary productivity and biomass could be attributed to the lower
precipitation estimated by CRU dataset. Past literature reported that reduction in precipitation can cause soil water
stress leading to reduction in stomatal conductance and reduction in leaf area (Konings et al., 2017; Ondier et al.,
433     2021).

**5 Conclusion**
The results of this study suggest that HKH will act as a net sink of C under both strong and weak scenarios of future
climate change. There was relatively good correspondence between the model and complimentary satellite-based
estimates of biomass and primary productivity. However, it is important to note that as long as obtainability and access
of meteorological data at a regional level and in-situ validation data such as eddy covariance measurements and long-
term ecological field assessments remain scarce, it can be expected the representativity of vegetation carbon and
vegetation productivity estimates for HKH to remain hard to evaluate definitively. The LPJ-GUESS simulations
revealed that the NBP is projected to be higher in future scenarios than in the historical period, assuming that the
LULCC does not increase dramatically. Furthermore VegC storage spatial and temporal analysis suggest that, for the
RCP8.5 scenario, the CMIP5 climate model produces, on average, a slightly higher VegC compared to the RCP2.6
attributing to the $CO_2$ fertilization effect in both PNV and LU simulations. Vegetation fluxes can help to analyse the
carbon storage patterns, however further studies are required to assess the effects of climatic changes and
anthropogenic activities on the fragile ecosystems of the HKH for the establishment of policies to improve the
livelihood of the local population and the overall carbon balance in the region.

**Acknowledgements.** This work was supported by NUST Research Grant for MS students.
**Data/Code Availability.** Data used in this study is available on the 4TU.ResearchData.
**Conflict of Interest.** There is no conflict of interest.

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
