# Peer review of "Climate Change Projections of Terrestrial Primary Productivity over the Hindu Kush Himalayan Forests"

_Earth System Dynamics, 2020_

## Referee Comment (RC1) · Anonymous Referee #1 · 30 Dec 2020

This manuscript entitled "Climate Change Projections of Terrestrial Primary Productivity over the Hindu Kush Himalayan Forests" used LPJ-GUESS terrestrial dynamic global vegetation model (DGVM) to analyse the ecosystem productivity and landuse change over the Hindu Kush mountain range. The region is quite under studied given its importance for the future climate sustainability. Thus the research topic is of high relevance to Earth System Dynamics, but as presented the analysis is shallow. I have pointed out a few points where the authors can improve their work. In summary, a big effort is needed by the authors themselves to present/explain the plots.

Specific comments:

Line 28: better to cite https://www.esrl.noaa.gov/gmd/ccgg/trends/gl_gr.html; what is 'presently'? during 2000-2019 the rate was above 2 ppm/yr. please give credit to the measurement people on such occasions

Line 50: remove "since 1960"

Figure 2: how some parts are appearing brown in the modelled VegC? how is the quality/accuracy of the GeoCarbon product over the HK region? Is there a product evaluation analysis?

Line 168: I agree with your assessment, but I see something systematic that the model using CCSM4 produce maximum Veg-C, and that using MPI is intermediate and lowest for IPSL. Please discuss the details, e.g., the with respect to the model drivers

Line 182: you did not define tundra in section 2.1? where are those located

Line 190: better to say 1-sigma, if true?

Figure 4: can this plot of NPP be reconciled with the VegC in fig.3? or VegC are a result of cumulative NPP over a longer period of time? need some discussions

NOT ENOUGH JUSTIFICATION TO WRITE ONE SENTENCE PER BIG FIGURES, Fig., 4 - Fig. 7. please discuss details if there is something interesting for the readers. else you need to delete most of these

Figure 9: It is a bit suspicious result that NBP increased during ~1960-2020 while the natural ecosystem is replaced by pasture. I do not know if this is an artifact of the CCSM4 meteorology or something else. at this point it should be nice to have a simulation case using CRU meteorology as submitted to the global carbon project using the same model. we need a good evaluation of the models for historical period and then discuss the projections, e.g., what you are showing in Fig. 10 in particular

---

## Referee Comment (RC2) · Anonymous Referee #2 · 5 Jan 2021

An interesting study focusing on a region expected to be sensitive to climate change. The study takes advantage of existing global model experiments from a previous study with a more detailed regional assessment. The manuscript is generally well written and presented but unfortunately there are key gaps in the detailed assessment necessary to reach the conclusions and assessments.

The authors are taking advantage of an existing ensemble so it is with reluctance that I suggest additional runs are required to attribute the drivers of changes in nbp. I would recommend following the C4MIP/LUMIP/TRENDY protocol of runs fixing components. In particular additional experiments with: Fixed PI or PD land-use Fixed PI CO2 con-

centrations. If not, more detailed assessment of the single runs is certainly required. In my assessment major revisions are required.

Major comments: 1. The climate model data is downscaled and bias corrected to a half-degree resolution using the CRU TS3.0 data. However, observations for the region are very sparse with considerable uncertainty in precipitation and other important fields. The interpolation of data to higher resolution elevation data is also potentially important and a possible advantage of this study. What confidence do you have in the application of the bias correction approach to a region of complex topography and sparse observations? How does the baseline CRU data compare to other observational estimates?

2. Climate uncertainty – the HKH is a region of high uncertainty in future climate response. For instance, there is uncertainty in the sign of change in western disturbances and monsoon affecting the HKH region. How do the available GCMs sample this uncertainty? It would be useful to see how representative these models are.

3. Detailed assessment of the components and drivers of changes in nbp and its components is generally missing. Analysis of the main results is generally thin.

Minor comments: Figure 1: White appears both within the HKH region and the rest of the region. What does it represent?

Spinup – can you confirm nbp is zero over the region at the end of the spin-up period? Are the PFT fractions prescribed or dynamically spun-up?

Figure 2: What period is covered here?

3.1 Can you explain why there are differences in the historical BC period?

3.2 What confidence do you have in the MODIS data set?

3.1/3.2 A key landclass of concern is EBF. It's not clear whether you are prescribing the vegetation cover or simulating interactively. Are there are insights here? It would be

useful to have some assessment of the PFTcover if it is dynamic particularly in regard to the application of the lans-use data.

3.3 You suggest land-use change fLuc is the cause of the decline in nbp but I miss any analysis of the nbp components that would justify this basis. It would be very useful to plot and analyse time series of the components. You mention crops and pasture but not how they are harvested and grazed. There is also no assessment of soil carbon and respiration which is a component of nbp.

3.3 Units are surely incorrect: 'The total VegC (averaged for all models) was estimated to be 7400 kg C m-2 by 1950'

---

## Author Comment (AC1) · 19 Mar 2021

We are grateful to reviewer for their corrections and comments. Kindly find the response to the comments below.

1) Major comments: 1. The climate model data is downscaled and bias corrected to a half-degree resolution using the CRU TS3.0 data. However, observations for the region are very sparse with considerable uncertainty in precipitation and other important fields. The interpolation of data to higher resolution elevation data is also potentially important and a possible advantage of this study. What confidence do you have in

the application of the bias correction approach to a region of complex topography and sparse observations? How does the baseline CRU data compare to other observational estimates?

Reply: Bias-correction is necessary because non-negligible biases in climate models can lead to unrealistic baseline ecosystem properties when fed through a vegetation model such as the LPJ-GUESS model used here. We agree that a high-resolution regional climate dataset, tied to ample observations, would provide improved confidence in the result. Furthermore, in applying a global model set-up, we are making an assessment of how well the HKH region is represented in the types of model simulations most commonly available for this region.

2) Climate uncertainty – the HKH is a region of high uncertainty in future climate response. For instance, there is uncertainty in the sign of change in western disturbances and monsoon affecting the HKH region. How do the available GCMs sample this uncertainty? It would be useful to see how representative these models are.

Reply: In the revised manuscript we will include plots of the temperature and precipitation anomalies for the region from a wider range of CMIP5 GCMs, situating the chosen GCMs here within that ensemble.

3) 3. Detailed assessment of the components and drivers of changes in nbp and its components is generally missing. Analysis of the main results is generally thin.

Reply: Components of NBP are shown below. Will be discussed in the manuscript. In LPJ-GUESS, the main components and drivers of changes consist of NBP consist of soil heterotrophic respiration, wildfire emission and vegetation NPP (Veg+Est). The time series have been changed to 1851-1880, 1986-2015 and 2071-2100 (RCP2.6 and RCP8.5). The attachments include spatial maps of average flux of soil (figure 1), fire (figure 2), NPP (figure 3) and NBP (figure 4) of HKH region respectively. The future RCP8.5 show a higher soil flux with mean value of 0.51 kg C m-2. Furthermore most of the flux values are the concentrated in the western part of HKH with an average

value of 0.68 kg C m-2 in RCP8.5. LPJ-GUESS simulations, show a negative NPP indicating decomposition or respiration is overpowered carbon absorption; more carbon was released to the atmosphere than the plants took in. Furthermore in the revised manuscript, additional graphs on the basis of these figures (spatial and temporal) will be added relating to drivers of NBP of HKH region according to high and low elevation and land use cover.

4) Minor comments: Figure 1: White appears both within the HKH region and the rest of the region. What does it represent?

Reply: White area within the HKH boundary represents area of barren, urban and water.

5) Spinup – can you confirm nbp is zero over the region at the end of the spin-up period? Are the PFT fractions prescribed or dynamically spun-up?

Reply: The relative prevalence of the different PFTs varies dynamically in response to the climate (temperature, precipitation, incoming shortwave radiation) and [CO2] forcing and the evolution (Ahlstrom). NBP averaged over the period 1851-1880 is 0.0034 kg C m-2 yr-1, 0.0058 kg C m-2 yr-1 and 0.001 kg C m-2 yr-1 models in simulations forced by IPSL-CM5A-MR, MPI-ESM-LR and CCSM4, respectively. This is well within expectations for a lack of trend. PFT fractions are dynamically spun-up on natural areas and prescribed on agricultural areas. We will add this information in the revised manuscript.

6) Figure 2: What period is covered here?

Reply: The period covered for GEOCARBON dataset is 2000 and for LPJ-GUESS is 1990-2015. However the LPJ-GUESS year for this figure will be changed to 1986-2015.

7) 3.1 Can you explain why there are differences in the historical BC period?

Reply: Vegetation carbon is different in the historical period when forced by different GCMs this is because the GCMs simulate slightly different climate variability for this

period. The bias correction used alters means, but not variability.

8)3.2 What confidence do you have in the MODIS data set?

Reply: The MODIS dataset is a well-established independent method to estimate NPP and GPP at these scales. We use it as an independent comparison for our results and do not suggest that it is a truth.

9) 3.1/3.2 A key landclass of concern is EBF. It's not clear whether you are prescribing the vegetation cover or simulating interactively. Are there are insights here? It would be useful to have some assessment of the PFTcover if it is dynamic particularly in regard to the application of the land-use data.

Reply: In all simulations, PFT cover was prescribed as grassland on pasture and agricultural grid-cell fractions, as specified by Hurtt et al. (2011). On all other areas the PFT cover was allowed to vary dynamically, as simulated by LPJ-GUESS. The land cover classification of MOD12Q1, was used in order to assess how variables (such as Veg, primary productivity) changed in different time periods. Land cover classification was not prescribed or simulated. Furthermore in the revised manuscript, an additional figure of LPJ-GUESS capturing the PFT distribution in the MODIS data will be added.

10) 3.3 You suggest land-use change fLuc is the cause of the decline in nbp but I miss any analysis of the nbp components that would justify this basis. It would be very useful to plot and analyse time series of the components. You mention crops and pasture but not how they are harvested and grazed. There is also no assessment of soil carbon and respiration which is a component of nbp.

Reply: Carbon fluxes due to crop and pasture harvest and grazing were not considered in these simulations, but are an important consideration for future work.

11) 3.3 Units are surely incorrect: 'The total VegC (averaged for all models) was estimated to be 7400 kg C m-2 by 1950'

Reply: The values and units will be updated.

[Figure]

**Fig. 1.** LPJ-GUESS simulated distribution by CCSM4 of Soil Flux in HKH region under a) past period (1851-1880) b) present period (1986-2015) and future scenario under c) RCP2.6 scenario and d) RCP8.5.

[Figure]

**Fig. 2.** LPJ-GUESS simulated distribution by CCSM4 of Fire Flux in HKH region under a) past period (1851-1880) b) present period (1986-2015) and future scenario under c) RCP2.6 scenario and d) RCP8.5.

[Figure]

**Fig. 3.** LPJ-GUESS simulated distribution by CCSM4 of NPP in HKH region under a) past period (1851-1880) b) present period (1986-2015) and future scenario under c) RCP2.6 scenario and d) RCP8.5.

[Figure]

**Net Biome Productivity (CCSM4)**

**Fig. 4.** LPJ-GUESS simulated distribution by CCSM4 on NBP in HKH region under a) past period (1851-1880) b) present period (1986-2015) and future scenario under c) RCP2.6 scenario and d) RCP8.5.

---

## Author Comment (AC2) · 19 Mar 2021

We are grateful to reviewer for their corrections and comments. Kindly find the response to the comments below.

1) Line 28: better to cite https://www.esrl.noaa.gov/gmd/ccgg/trends/gl_gr.html; what is 'presently'? during 2000-2019 the rate was above 2 ppm/yr. please give credit to the measurement people on such occasions.

Reply: Change has been made.
2) Line 50: remove "since 1960"

Reply: Acknowledged

3) Figure 2: how some parts are appearing brown in the modelled VegC? how is the quality/accuracy of the GeoCarbon product over the HKH region? Is there a product evaluation analysis?

Reply: The LPJ-GUESS shows some values for the upper part of HKH (such as 0, 0.2, 0.5 etc). However since there is low vegetation present in this area the GEOCARBON does not show any forest inventory based data. Product evaluation is not present.

4) Line 168: I agree with your assessment, but I see something systematic that the model using CCSM4 produce maximum Veg-C, and that using MPI is intermediate and lowest for IPSL. Please discuss the details, e.g., the with respect to the model drivers

Reply: Components of NBP are shown below (at the end). Will be discussed in the revised manuscript. In LPJ-GUESS, the main components and drivers of changes consist of NBP consist of soil heterotrophic respiration, wildfire emission and vegetation NPP (Veg+Est). The time series have been changed to 1851-1880, 1986-2015 and 2071-2100 (RCP2.6 and RCP8.5). The attachments include spatial maps of average flux of soil (figure 1), fire (figure 2), NPP (figure 3) and NBP (figure 4) of HKH region respectively. The future RCP8.5 show a higher soil flux with mean value of 0.51 kg C m-2. Furthermore most of the flux values are the concentrated in the western part of HKH with an average value of 0.68 kg C m-2 in RCP8.5. LPJ-GUESS simulations, show a negative NPP indicating decomposition or respiration is overpowered carbon absorption; more carbon was released to the atmosphere than the plants took in. Furthermore in the revised manuscript, additional graphs on the basis of these figures (spatial and temporal) will be added relating to drivers of NBP of HKH region according to high and low elevation and land use cover.

5) Line 182: you did not define tundra in section 2.1? where are those located

Reply: In this study, the land cover of MOD12Q1 has been utilized. The sentence has been modified and barren area have been incorporated instead of tundra.

6) Line 190: better to say 1-sigma, if true?

Reply: We prefer to use the unambiguous term "standard error".

7) Figure 4: can this plot of NPP be reconciled with the VegC in fig.3? or VegC are a result of cumulative NPP over a longer period of time? need some discussions

Reply: Figure 4 cannot be reconciled with figure 3 as both figures time period is different. Figure 4 time line period is 2000-2010 whereas Figure 3 shows the time line of a single year 2000. However, Figure 7 does show the GPP and NPP according to landcover classes similarly as Figure 3.

8) NOT ENOUGH JUSTIFICATION TO WRITE ONE SENTENCE PER BIG FIGURES, Fig., 4 - Fig. 7. please discuss details if there is something interesting for the readers. else you need to delete most of these

Reply: The text in the manuscript will be expanded in the future revised manuscript related to the figures focusing on the in depth analysis of the variables discussed.

Figure 9: It is a bit suspicious result that NBP increased during âĹij1960-2020 while the natural ecosystem is replaced by pasture. I do not know if this is an artifact of the CCSM4 meteorology or something else. at this point it should be nice to have a simulation case using CRU meteorology as submitted to the global carbon project using the same model. we need a good evaluation of the models for historical period and then discuss the projections, e.g., what you are showing in Fig. 10 in particular

Reply: For clarification figure 10, shows the NBP in three different time periods. However the time period will be changed to 1851-1880, 1986-2015, 2071-2100 (30 year interval) in the revised manuscript. Plots of the temperature and precipitation anomalies for the region from a wider range of CMIP5 GCMs, situating the chosen GCMs here within that ensemble will be included in the revised manuscript.

[Figure]

**Soil Flux (CCSM4)**

(a) 1851-1880      (b) 1986-2015

(c) 2071-2100 (RCP2.6)      (d) 2071-2100 (RCP8.5)

**Fig. 1.** LPJ-GUESS simulated distribution by CCSM4 of Soil Flux in HKH region under a) past period (1851-1880) b) present period (1986-2015) and future scenario under c) RCP2.6 scenario and d) RCP8.5.

[Figure]

**Fig. 2.** LPJ-GUESS simulated distribution by CCSM4 of Fire Flux in HKH region under a) past period (1851-1880) b) present period (1986-2015) and future scenario under c) RCP2.6 scenario and d) RCP8.5.

[Figure]

**NPP (Est+Veg) (CCSM4)**

(a) 1851-1880  (b) 1986-2015

(c) 2071-2100 (RCP2.6)  (d) 2071-2100 (RCP8.5)

**Fig. 3.** LPJ-GUESS simulated distribution by CCSM4 of NPP in HKH region under a) past period (1851-1880) b) present period (1986-2015) and future scenario under c) RCP2.6 scenario and d) RCP8.5.

[Figure]

**Net Biome Productivity (CCSM4)**

**Fig. 4.** LPJ-GUESS simulated distribution by CCSM4 on NBP in HKH region under a) past pe-
riod (1851-1880) b) present period (1986-2015) and future scenario under c) RCP2.6 scenario
and d) RCP8.5.